# Region-Specific Gene Expression Changes Associated with Oleoylethanolamide-Induced Attenuation of Alcohol Self-Administration

**DOI:** 10.3390/ijms25169002

**Published:** 2024-08-19

**Authors:** Macarena González-Portilla, Sandra Montagud-Romero, Susana Mellado, Fernando Rodríguez de Fonseca, María Pascual, Marta Rodríguez-Arias

**Affiliations:** 1Department of Psychobiology, Faculty of Psychology, Universitat de València, Avda. Blasco Ibáñez 21, 46010 Valencia, Spain; macarena.gonzalez@uv.es (M.G.-P.); sandra.montagud@uv.es (S.M.-R.); 2Department of Physiology, School of Medicine, Universitat de Valencia, Avda. Blasco Ibáñez 15, 46010 Valencia, Spain; susana.mellado@uv.es (S.M.); maria.pascual@uv.es (M.P.); 3Mental Health Clinical Management Unit, Institute of Biomedical Research of Malaga-IBIMA, Regional University Hospital of Málaga, 29010 Málaga, Spain; fernando.rodriguez@ibima.eu; 4Atención Primaria, Cronicidad y Promoción de la Salud, Red de Investigación en Atención Primaria de Adicciones (RIAPAD) Rd21/0009/0005/0003, Valencia, Spain

**Keywords:** alcohol, oleoylethanolamide, dopamine, cannabinoid, gene expression, immune, BDNF

## Abstract

Oleoylethanolamide (OEA) is a lipid with anti-inflammatory activity that modulates multiple reward-related behaviors. Previous studies have shown that OEA treatment reduces alcohol self-administration (SA) while inhibiting alcohol-induced inflammatory signaling. Nevertheless, the specific mechanisms that OEA targets to achieve these effects have not been widely explored. Here, we tested the effects of OEA treatment during alcohol SA, extinction or previous to cue-induced reinstatement of alcohol seeking. In addition, we measured gene expression changes in the striatum and hippocampus of relevant receptors for alcohol consumption (Drd1, Drd2, Cnr1, Oprm) as well as immune-related proteins (Il-6, Il-1β, Tlr4) and the brain-derived neurotrophic factor (Bdnf). Our results confirmed that when administered contingently, systemic OEA administration reduced alcohol SA and attenuated cue-induced reinstatement. Interestingly, we also observed that OEA treatment reduced the number of sessions needed for the extinction of alcohol seeking. Biochemical analyses showed that OEA induced gene expression changes in dopamine and cannabinoid receptors in the striatum and hippocampus. In addition, OEA treatment modulated the long-term immune response and increased Bdnf expression. These results suggest that boosting OEA levels may be an effective strategy for reducing alcohol SA and preventing relapse.

## 1. Introduction

Alcohol use disorder (AUD) is the most prevalent substance use disorder, which has a significant impact on public health and society [1]. Chronic alcohol exposure profoundly affects various systems in the human body, including the cardiovascular, gastrointestinal, and nervous systems. AUD is characterized by significant dysregulation in the brain’s motivational circuits, driven by exaggerated incentive salience, habit formation, deficits in reward processing, increased stress, and impaired executive function [2,3,4,5]. These disruptions are primarily due to alcohol’s effects on synaptic function, impacting multiple neurotransmitter systems, such as serotonergic (5-HT), dopaminergic (DA), gamma-aminobutyric acid (GABA), glutamatergic (Glu), acetylcholinergic (ACh), and opioid systems [2,3,4,5]. Additionally, alcohol consumption is associated with inflammatory states, significantly influencing alcohol-related behaviors. Chronic alcohol intake can exacerbate inflammation, creating a vicious cycle in which inflammation further promotes alcohol consumption. This relationship is mediated through various mechanisms, including the activation of Toll-like receptors (TLRs) and their downstream pathways, in both peripheral cells (e.g., monocytes) and central cells (e.g., microglia) [5]. Furthermore, alcohol-induced gut dysbiosis results in the release of endotoxins from a compromised intestinal barrier, which circulate to the brain and other organs, triggering systemic and central inflammation. These inflammatory processes can alter neurochemical pathways, impacting mood, cognition, and reward processing [6,7]. Consequently, individuals with heightened inflammatory states may experience increased alcohol cravings and consumption, thereby exacerbating the risk of AUD [7].

Currently, there are several treatments available for AUD. Regulatory agencies have approved several medications, including disulfiram, naltrexone, and acamprosate [3]. The available pharmacological therapies have modest effects on reducing the risk of relapse and do not target the negative emotional state related to alcohol abstinence [8]. Furthermore, there are several promising off-label pharmacotherapy options, such as nalmefene, baclofen, and topiramate. Despite the high prevalence, mortality, and economic impact of AUD, these medications remain under-prescribed. Studies indicate that only a small number of individuals in need of treatment receive even a single prescription for any approved medication. This is likely due to barriers such as limited access, financial constraints, or inadequate health insurance coverage [3,4].

In recent years, lipid-signaling molecules have emerged as important modulators of reward-related behavior [9]. Oleoylethanolamide (OEA) is an endogenous molecule derived from oleic acid, which functions as a satiety signal [10,11]. OEA has a common biosynthetic and degradation pathway with other *N*-acylethanolamines (NAEs), anandamide (AEA), and palmitoylethanolamide (PEA) [12]. All NAEs are synthesized on demand from lipid precursors, and their deactivation depends mainly on hydrolysis by the enzyme fatty acid amide hydrolase (FAAH). OEA levels fluctuate during the day according mainly to nutritional status. In this regard, OEA levels increase after feeding and especially upon the intake of lipids [11]. As a high-caloric substance, alcohol has also been proved to rapidly stimulate OEA production in enterocytes of the small intestine [13]. Recently, a clinical study demonstrated that levels of NAEs, including OEA, are altered in patients with AUD. The plasma concentration of OEA was found to be elevated in alcohol-dependent patients during abstinence. Additionally, OEA concentrations were negatively correlated with the duration of alcohol abstinence [14,15].

In preclinical studies, OEA has been shown to affect the motivation for voluntary alcohol intake using animal models of both chronic and acute alcohol drinking. Using the self-administration (SA) and the two-bottle choice paradigms, OEA administration reduced alcohol drinking at 5, 10, and 20 mg/kg doses [13,16]. OEA was also effective in blocking the cue-induced reinstatement of alcohol-seeking behavior [13]. In addition, OEA has been proved to reduce the negative effect associated with alcohol withdrawal [13]. These beneficial effects have been argued to depend on peripheral mechanisms since direct OEA infusion in the nucleus accumbens does not affect alcohol consumption [13]. In this context, the endogenous compound oleoylethanolamide (OEA) has been shown to play a significant role in modulating alcohol-related behavior through its effects on the immune response. Research suggests that the exogenous administration of OEA may act on peripheral receptors and signal to the central nervous system, reducing alcohol consumption. OEA interacts with peroxisome proliferator-activated receptors (PPAR-α) expressed in immune cells, altering the transcription of target genes involved in the immune response [17]. Numerous studies have demonstrated that OEA prevents inflammatory signaling induced by alcohol exposure in several brain structures [18]. As a consequence, we determined gene expression changes in the striatum and hippocampus, brain structures critically involved in the pathophysiology of drug abuse (striatum adjusts reward processing, while the hippocampus plays a crucial role in learning and memory processes). These changes include alterations in the immune-related genes, such as interleukin-1β (Il-1β), interleukin-6 (Il-6), Toll-like receptor-4 (Tlr4), and brain-derived neurotrophic factor (Bdnf), as well as variations in dopamine receptor D1 (Drd1), dopamine receptor D2 (Drd2), cannabinoid receptor 1 (Cnr1), and mu-type opioid receptor (Oprm). Previous studies in our laboratory found that OEA altered dopaminergic and cannabinoid gene expression in these structures [19]. These findings highlight the complex interplay between inflammatory states, neurochemical pathways, and alcohol-related behaviors, underscoring the importance of targeting inflammatory processes in the treatment of alcohol use disorders.

To date, the effects of OEA on central gene expression associated with alcohol consumption have not been described. To further characterize the effects of OEA on voluntary alcohol drinking, we also compared different OEA administration schedules at different timepoints during the alcohol SA procedure.

## 2. Results

### 2.1. Bodyweight

Given the reported effects of OEA on feeding behavior, we obtained a weekly measure of mice bodyweight. The ANOVA revealed a significant effect of the variable week [F(8,336) = 98.819 (*p* < 0.001)], as mice exhibited an increased bodyweight across weeks compared to the first measurement (*p* < 0.001) (Figure 1). No differences were found between groups.

### 2.2. DID and Oral Alcohol SA

The ANOVA for the DID did not reveal any differences among groups. Since the pharmacological treatment began after the DID procedure, the lack of differences was expected.

The ANOVA for ethanol consumption during the FR1 schedule revealed a significant effect of the variable treatment [F(1,44) = 4091; *p* = 0.049]. A lower EtOH consumption was observed in OEA-FR1 mice compared to the CTRL, OEA-EXT, and OEA-REINST mice (*p* = 0.049) (Figure 2).

Regarding active responses, the ANOVA showed an effect of the variable treatment [F(1,44) = 4827; *p* = 0.033]. Mice in the OEA-FR1 group performed fewer active responses compared to the CTRL, OEA-EXT, and OEA-REINST mice (*p* = 0.033).

During the PR, the ANOVA revealed a significant effect of the variable “Treatment” [F(3,42) = 57.957 (*p* < 0.001). Post hoc comparisons showed that alcohol consumption was decreased in OEA-FR1 mice compared to CTRL (*p* = 0.05), OEA-EXT (*p* = 0.05), and OEA-REINST (*p* = 0.01) groups.

With regard to the time required to extinguish alcohol SA, the OEA-EXT group required a mean number of 3.1 sessions, while the CTRL, OEA-FR1, and OEA-REINST groups required a mean number of 4.4, 4, and 4.2 sessions, respectively (Figure 3A). The Kaplan–Meier analysis revealed that the OEA-EXT group required significantly fewer sessions than the CTRL (χ^2^ = 4.380; *p* = 0.036) and OEA-REINST (χ^2^ = 5.253; *p* = 0.022) to meet the extinction criteria.

Finally, we tested the effects of OEA on drug-seeking behavior induced by contextual cues (Figure 3B). The ANOVA revealed an effect of the variable day [F(1,41) = 16,455; *p* = 0.001] and the interaction day x treatment [F(3,41) = 3147; *p* = 0.037]. The active responses induced by the alcohol-associated stimuli on the reinstatement test were significantly increased with respect to the last three days of extinction in the CTRL and OEA-EXT groups (*p* < 0.05 and *p* < 0.001, respectively). No differences in the number of active responses were observed for mice in the OEA-FR1 and OEA-REINST groups.

### 2.3. OEA Affects Gene Expression in the Striatum and Hippocampus

The ANOVA revealed an effect of the variable treatment for striatal Drd1 [F(3,28) = 4595; *p* = 0.01], cannabinoid receptor-1 gene (Cnr1) [F(3,28) = 3539; *p* = 0.027], and Oprm1 [F(3,28) = 3352; *p* = 0.033] gene expression. OEA administration in the OEA-FR1 group significantly decreased Drd1 mRNA expression compared to the CTRL (*p* = 0.01) and the OEA-REINST group (*p* = 0.052). Similarly, a decrease in Cnr1 mRNA expression was observed in the striatum of OEA-FR1 mice compared to OEA-REINST mice (*p* = 0.024). In addition, we found increased levels of Oprm1 gene expression in the striatum in OEA-EXT mice compared to CTRL mice (*p* = 0.037) (Figure 4A).

Regarding the expression of inflammatory genes in the striatum, the ANOVA revealed an effect of the variable treatment for Il-1β [F(3,28) = 367; *p* = 0.024] and Tlr4 [F(3,28) = 6873; *p* = 0.001]. We found increased mRNA expression of Il-1β in the striatum of OEA-FR1 mice compared to CTRL mice (*p* = 0.031). Similarly, the gene expression of Tlr4 was increased in the striatum in OEA-FR1 mice compared to CTRL (*p* = 0.42), OEA-EXT (*p* = 0.59), and OEA-REINST mice (*p* < 0.001) (Figure 4B).

Finally, the ANOVA for the striatal Bdnf gene expression showed an effect of the variable treatment [F(3,27) = 3922; *p* = 0.019]. Gene expression analyses showed that OEA-FR1 mice exhibited increased Bdnf mRNA levels compared to the CTRL mice (*p* < 0.30) (Figure 4B).

The ANOVA revealed an effect of the variable treatment for hippocampal Drd1 [F(3,28) = 5341; *p* = 0.005], Cnr1 [F(3,27) = 12,892; *p* < 0.001] and Oprm1 gene expression [F(3,28) = 4514; *p* = 0.011]. An increase in Drd1 gene expression was observed in the hippocampus of OEA-FR1 compared to OEA-EXT mice (*p* = 0.004). Also, an increase in hippocampal Cnr1 gene expression was observed in OEA-FR1 compared to the CTRL (*p* = 0.002), OEA-EXT (*p* < 0.001), and OEA-REINST groups (*p* = 0.10). In addition, Oprm1 gene expression was decreased in the hippocampus of OEA-FR1 mice compared to the CTRL (*p* = 0.40), OEA-EXT (*p* = 0.45), and OEA-REINST groups (*p* = 0.22) (Figure 5A).

Regarding the gene expression of inflammatory immune receptor, Tlr4, in the hippocampus, the ANOVA revealed an effect of the variable treatment [F(3,28) = 3884; *p* = 0.019]. The results also showed a downregulation in its mRNA levels in OEA-FR1 mice compared to OEA-EXT mice (Figure 5B).

Finally, the ANOVA for the hippocampal Bdnf gene expression showed an effect of the variable treatment [F(3,27) = 3217; *p* = 0.38]. These analyses showed that OEA-FR1 mice exhibited increased Bdnf mRNA levels compared to the CTRL mice (*p* = 0.38).

We performed a Pearson correlation analysis (Figure 6) between the total ethanol intake during the DID and FR1 procedures and the expression of the studied genes in the hippocampus and striatum. No significant correlations were found. Similarly, no significant correlations were observed between the number of effective responses during the progressive ratio or reinstatement test. However, we identified a significant positive Pearson correlation between the number of days required to achieve extinction and the hippocampal expression of Drd1 (r = 0.382, *p* < 0.031) and Cnr1 (r = 0.378, *p* < 0.036). This suggests that higher gene expression levels are associated with an increased number of days needed to extinguish operant ethanol self-administration. Additionally, a significant negative correlation was observed with Il-1β expression (r = −0.382, *p* < 0.031), indicating that higher Il-1β gene expression is associated with fewer days required to achieve extinction.

## 3. Discussion

Previous studies have identified OEA as a lipid that modulates responses to natural rewards, such as food, and to drugs of abuse. In this study, we examined the effects of OEA on alcohol self-administration (SA). We confirmed that acute OEA administration (10 mg/kg) reduces alcohol consumption and accelerates the extinction of alcohol-seeking behavior. Additionally, a single OEA dose blocks the cue-induced reinstatement of alcohol seeking. This study also explores the effects of OEA on the expression of relevant genes in the striatum and hippocampus, including those associated with alcohol-related behavior (Drd1, Drd2, Cnr1, Oprm), neuroinflammatory processes (Il-1β, Il-6, and Tlr4), and Bdnf.

As previously reported, we observed that acute OEA administration before each session reduces operant responses for alcohol SA and consumption [13]. Similar to the FR1 period, mice in the OEA-FR1 group exhibited decreased alcohol consumption during the PR session (Figure 2). Although not significant, a similar trend was observed in the breaking point values, suggesting a reduction in the motivation to obtain Alcohol A potential limitation of this study is the absence of blood ethanol concentration measurements, which could have been correlated with ethanol-induced reinforcement.

One of the most notable findings of this study was the effect of OEA on the extinction of alcohol-associated memories. Considering that OEA reduces the severity of alcohol withdrawal symptoms [13,20], we examined its effects during the extinction period (OEA-EXT group), where neither alcohol nor alcohol-associated cues were present following an active response. Remarkably, OEA treatment before each extinction session reduced the time required to extinguish alcohol-seeking behavior (Figure 3A). This is the first study to demonstrate OEA’s effect on the extinction of alcohol-seeking behavior. These findings align with human studies showing increased serum OEA levels in AUD patients during alcohol withdrawal [15]. OEA may enhance cognitive functions, thereby facilitating the learning mechanisms involved in extinction. Post-training OEA administration improves retention in inhibitory avoidance and Morris water maze tasks in rats [21]. Additionally, recent research reported improved cognitive functions in a mouse model of Wernicke-Korsakoff syndrome [22]. Thus, OEA may expedite the consolidation of extinction learning while alleviating the aversive symptoms of alcohol deprivation, reducing alcohol-related motivational memories.

Consistent with a previous study, we also observed that a single dose was effective in blocking the cue-induced reinstatement of alcohol seeking (Figure 3B) [13]. It is possible that OEA improves the reward deficit induced by alcohol SA and ameliorates craving and depressive-like behavior, which prevent reinstatement. In addition, OEA-FR1 did not reinstate cue-induced alcohol seeking, highlighting the decreased learning established during the FR1 period under the effect of OEA. To gain better insight into the neurotransmission system involved in OEA effects on alcohol-related behaviors, we evaluated the transcriptional regulation in two brain regions of the reward system, the striatum and hippocampus.

Changes in gene expression, both in the striatum and hippocampus, were mainly observed in the OEA-FR1 group, following the administration of 10 doses of OEA during the FR1 phase of oral ethanol self-administration. It is important to remark that at the point of tissue collection, the time from the OEA treatment as well as the number of doses between groups differed. It is important to note that samples were taken at the end of the entire SA procedure, so 7 days had passed since the last administration of OEA in the case of the OEA-FR1 group, indicating that the changes induced by this administration regimen persist over time.

In humans, postmortem studies show that the number of binding sites of dopamine DRD1 and CNR1 is increased in the striatum of individuals with a history of AUD [23]. Our results showed that Drd1 gene expression was decreased in the striatum of OEA-FR1 mice. Consistent with these results, striatal Drd1 gene expression was decreased in mice that received OEA during each session of cocaine-induced conditioned place preference compared to the vehicle-treated mice [19]. Therefore, our results show that OEA altered Drd1 gene expression in the striatum of mice that received OEA contingently to alcohol SA, which may be related to the reduction in alcohol consumption (Figure 4A). However, OEA administration during FR1 did not change Drd1 gene expression in the hippocampus with respect to the control group, although there was a tendency to increase gene expression (Figure 5A).

It has been demonstrated that the dopaminergic system is able to interact with cannabinoid signaling, modulating inputs to the striatum [24]. In this sense, it is possible that OEA reduces alcohol consumption by acting on these two neurotransmission systems in reward-related areas. In the striatum, decreased Cnr1 gene expression was detected in OEA-FR1 mice only with respect to the OEA-REINST group. At 24 h after injection, OEA increased Cnr1 gene expression, possibly due to competition for the catalytic function of FAAH [25]. Multiple studies have consistently shown that the genetic or pharmacological blockade of Cnr1 reduces most alcohol-related behaviors such as alcohol drinking and reinstatement in the operant paradigm [26,27,28]. According to these results, we suggest that multiple doses in OEA-FR1 mice affect the dopaminergic and endocannabinoid system, diminishing the rewarding properties of alcohol and resulting in decreased alcohol intake. Again, contrary results were obtained in the hippocampus. The molecular changes involving the dopaminergic and endocannabinoid systems, including downregulation of the CB1 receptor, are complex and frequently region-specific [29,30]. Also, lipidomic analyses in mice trained for alcohol SA show heterogeneous lipid profiles, including OEA, in multiple brain areas [31]. The expression of the Drd1 and Cnr1 genes positively correlates with the number of days required to extinguish self-administration. The expression of both genes is significantly reduced in the group that received OEA during extinction and required far fewer extinction sessions, which explains this correlation.

Due to the implication of the opioid system in alcohol reward, we also examined Oprm1 gene expression. In the striatum, the OEA-EXT group showed increased gene expression of Oprm1 compared to the CTRL group (Figure 4A). The specific cause for this upregulation is unclear, although it may be attributable to the blunting effect of OEA on the negative emotional state resulting from alcohol abstinence, which affects opioid signaling [32]. Differently from the striatum, OEA-FR1 mice exhibited a downregulation of Oprm1 in the hippocampus (Figure 5A). In fact, several studies have proved that mice lacking the Oprm1 gene exhibit a lower preference for alcohol and reduced SA compared to wild-type mice [33].

Another potential mechanism by which OEA may be exerting its effects is by modulating the alcohol-induced immune signaling. Multiple studies proved that OEA reduces the release of proinflammatory factors such as IL-1β or IL-6 [34,35]. An important signaling cascade of alcohol-induced immune response is TLR4 receptors [36]. Recent reports have proved the protective effects of OEA on acute and chronic activation of TLR4 by drug exposure [22]. In our study, we assessed transcription changes in several inflammatory-related genes after alcohol SA finished (Figure 4B and Figure 5B), since we aimed to detect lasting immune signatures associated with alcohol abstinence and craving. In the striatum, we observed an upregulation in immune genes (e.g., Il-1β and Tlr4) in OEA-FR1 mice compared to the CTRL group. In the hippocampus, the upregulation of Tlr4 expression was specific to the OEA-EXT group compared to OEA-FR1 mice. Since the peak inflammatory signaling occurred concomitantly to alcohol consumption, we presume that during the FR1 period, the immune response was attenuated in the OEA-FR1 mice by the OEA anti-inflammatory activity. In this sense, it is possible that the detected increase in Tlr4 results from a rebound effect. Given the timepoint when the neuroimmune gene transcript was obtained, this increase in the OEA-FR1 group likely reflects homeostatic interactions between different immune systems. The negative correlation observed between the days required to achieve extinction of SA and the expression of the Il-1β gene in the hippocampus is explained by the increase in this gene expression in the OEA-Ext group, as this group experienced the fastest extinction.

Finally, we also examined the neuroprotective growth factor BDNF, which is an important regulator of activity-driven synaptic plasticity [37]. The highest number of BDNF receptors is found in the hippocampus [38]. Previous studies show that OEA treatment restores BDNF following cerebral ischemia [39] or after stress exposure [40]. In our study, we observed that all OEA-FR1 mice exhibited higher levels of Bdnf gene expression compared to the CTRL group in the striatum and hippocampus (Figure 4B and Figure 5B), in agreement with previous reports [41]. Multiple doses of exogenous OEA could buffer the alcohol-induced neuronal toxicity by enhancing Bdnf expression.

All in all, we propose that the observed gene expression alterations are associated with the OEA-induced reduction in alcohol SA, extinction of alcohol seeking, and attenuated alcohol reinstatement. Nevertheless, different central and peripheral mechanisms may be contributing to this effect. Additionally, it is important to remark that many of the changes in dopaminergic, cannabinoid receptor, and immune-related gene expression may be secondary to the OEA-induced reduction in alcohol consumption in the OEA-FR1 group.

## 4. Methods and Materials

### 4.1. Experimental Design

A total of 46 OF1-strain male mice arrived at our laboratory on postnatal day (PND) 42 (Charles River, France) and were housed in groups of 4 in standard plastic cages (27 × 27 × 14 cm) under constant temperature and a reverse 12 h light/dark cycle (lights on at 7:00 h). In preclinical research, OF1 mice have been used for ethanol self-administration studies due to their well-established genetic and behavioral characteristics that enhance experimental reproducibility [42,43]. Mice were provided food and water ad libitum, except during SA testing. All procedures were conducted in compliance with the guidelines of the European Council Directive 2010/63/EU regulating animal research and were approved by the local ethics committees of the University of Valencia (2019/VSC/PEA/0065). The experimental design is depicted in Figure 7.

### 4.2. Drugs

OEA (10 mg/kg, i.p.; synthesized as described in [9]) was dissolved in 5% Tween 80 in saline and injected 10 min before the corresponding timepoint according to the experimental condition. The doses were chosen according to previous studies in mice reporting effective therapeutic and anti-inflammatory effects [19,22,35].

### 4.3. Habituation to Alcohol

The Drinking in the Dark (DID) paradigm was employed as a pre-exposure procedure to habituate mice to ethanol before starting the oral SA. Based on the basic paradigm of Rhodes et al. (2005) [44], this voluntary drinking protocol consisted of two phases. On the first day, mice were moved from their home cage for two hours to habituate to the individual cages and drinking tubes. In the second phase of the protocol, mice received 3 days of 2 h/day access to 20% (*v*/*v*) ethanol solution starting 3 h after lights off. On day 4, the procedure extended for 4 h. After each DID session, mice returned to their home cage.

### 4.4. Oral Alcohol Self-Administration (SA)

This procedure is based on that employed by Navarrete et al. (2014) [45]. Voluntary oral alcohol SA administration was carried out in eight modular operant chambers (MED Associated Inc., Georgia, VT, USA) equipped with a chamber light, two nose poke holes, one fluid receptacle, one syringe pump, one stimulus light, and one buzzer. The chambers were placed inside sound-attenuated cubicles. Designated active nose pokes delivered 20 μL of fluid associated to a 0.5 s light cue and a 0.5 s buzzer beep, which was followed by a 6 s time-out period. Inactive nose pokes triggered no event. Software package (Cibertec, SA, Madrid, Spain) controlled stimulus and fluid delivery and recorded operant responses.

The procedure consisted of five phases: training, fixed ratio 1 (FR1), progressive ratio (PR) SA of 20% (*v*/*v*) ethanol solution, extinction, and reinstatement. Alcohol SA behavior was assessed by daily operant sessions of 60 min duration.

#### 4.4.1. Training (15 Days)

Mice were trained to poke the active hole for 20% (*v*/*v*) ethanol solution delivery (20 μL) under FR1. Inclusion criteria for the next step in the procedure comprised 60% discrimination for active, over inactive nose poke responding across the three last days of this phase. Mice that met these criteria were randomly divided into four experimental groups: CTRL, OEA-FR1, OEA-EXT, and OEA-REINST groups (n = 10/12).

#### 4.4.2. FR1 (10 Days)

The number of effective responses and 20% ethanol (*v*/*v*) consumption (μL) were measured under FR1 for 10 daily sessions. After each session, the alcohol solution that remained in the receptacle was collected and measured with a micropipette.

#### 4.4.3. PR (1 Day)

Mice undertook a single 2 h long PR session in which the response requirement necessary to obtain one reinforcement escalated according to the following series: 1-2-3-5-12-18-27-40-60-90-135-200-300-450-675-1000. Breaking point (BP) was defined as the highest number of nose pokes each mouse performed to earn one reinforcement. This value was used to quantify motivational strength.

#### 4.4.4. Extinction Sessions

All mice progressed to the extinction phase, which consisted of removal of the cue light, buzzer, and the reward (alcohol) delivery after a response on the active nose poke during the 60 min session.

The extinction phase continued until the average in each experimental group reached the criterion (at least 50% decrease in active nose poke responses for at least three consecutive days).

#### 4.4.5. Cue-Induced Reinstatement of Alcohol Seeking (1 Day)

Following the extinction phase, a 60 min single session of cue-induced reinstatement was performed. Effective responses on the active nose poke were followed by presentation of cue light and buzzer stimuli in the absence of alcohol delivery. For this session, fluid receptacle was primed with alcohol solution.

### 4.5. Tissue Sampling and Biochemical Analyses

Mice were sacrificed by cervical dislocation. The entire striatum and hippocampus were precisely dissected out using a coronal brain matrix, while taking appropriate measures to avoid mRNA contamination of other samples and RNase contamination. Tissue samples were stored at −80 °C until the qRT-PCR assay was performed.

#### RNA Isolation, Reverse Transcription, and Quantitative RT-PCR

Striatum and hippocampus were lysed in Tri-Reagent solution (Sigma-Aldrich, St. Louis, MO, USA), and total RNA was isolated according to the manufacturer’s instructions. Then, the mRNA was reverse transcribed by the High Capacity cDNA Reverse Transcription Kit (Applied Biosystems, Waltham, MA, USA). Amplification of the target and housekeeping genes was completed employing the AceQ ^®^ qPCR SYBR Green Master Mix (NeoBiotech, Nanterre, France) and TaqManTM Fast Advanced Master Mix (Applied Biosystems, MA, USA), following the manufacturer’s instructions in a QuantStudioTM 5 Real-Time PCR System (Applied Biosystems, MA, USA). The mRNA level of housekeeping genes (cyclophilin A and β-glucuronidase) was used as an internal control for the normalization of the analyzed genes. Experiments were performed in triplicate. All the RT-qPCR runs included non-template controls (NTCs), and melt curve analysis was also used to ensure that the assays produce a single, specific qPCR product. Quantification of expression (fold change) from the Cq data was calculated by the QuanStudioTM Design & Analysis Software version 2.7.0 (Applied Biosystems). Details of the nucleotide sequences and the assay codes of the used primers are detailed in the Appendix A).

### 4.6. Statistical Analyses

Data relating to bodyweight were analyzed by a repeated measure ANOVA with one between-subjects variable “Treatment” with 4 levels (CTRL, OEA-FR1, OEA-EXT, OEA-REINST) and a within variable “Weeks” with 9 levels (1–9).

For the SA, alcohol consumption and active nose poke during responses during FR1 were analyzed individually by a repeated measure ANOVA with one between-subjects variable, treatment (CTRL, OEA), and a within-subjects variable, days, with ten levels of FR1 schedule.

A one-way ANOVA with a between-subjects variable—treatment (CTRL, OEA-FR1, OEA-EXT and OEA-REINST)—was employed to analyze alcohol consumption and breaking point values during the PR session.

The time required for meeting the extinction criteria in each animal was analyzed by means of the Kaplan–Meier test with Breslow (generalized Wilcoxon) comparisons. This analysis is appropriated in scenarios where the objective is to estimate the time until an event of interest occurs (extinction criteria).

To test for the cue-induced reinstatement, the number of active nose poke responses was analyzed with a two-way ANOVA with one between-subjects variable—treatment, with two levels (CTRL, OEA)—and one within subjects variable with two levels (day: extinction and cue-induced reinstatement).

The gene expression data were analyzed by a one-way ANOVA with one between variable, treatment, with four levels (CTRL, OEA-FR1, OEA-EXT, OEA-REINST). Bonferroni post hoc tests were also analyzed. Statistical analyses were performed using SPSS Statistics v23. A significance level of *p* < 0.05 was adopted throughout the study. In addition, correlation analysis between ethanol intake during DID and oral SA and gene expression was performed using Pearson’s correlation coefficient (r).

## 5. Conclusions

The present findings support the role of OEA as a homeostatic mediator modulating multiple alcohol-related behaviors. Taken together, OEA administration reduced alcohol SA, accelerated the extinction of alcohol seeking, and attenuated cue-induced reinstatement. The results of the present study and our previous work indicate that OEA is the most effective option in decreasing drug reward when administered contingently. Biochemical analyses suggest that OEA administration affects the gene expression of dopaminergic, cannabinoid, opioid, and immune-related factors in the striatum and hippocampus, key brain areas for reward processing. The impact of OEA on gene expression was heterogeneous, suggesting that both the quantity of doses and the timing of administration are crucial factors. A notable limitation of this study is the exclusive use of male mice. It is well established that there are significant gender differences in AUD, with complex interactions among genetic, epigenetic, hormonal, and environmental factors contributing to these disparities [46]. Furthermore, aspects such as the higher prevalence of mental disorders, including depression, associated with AUD in women, demand additional studies to develop treatments tailored specifically for this demographic [47]. Not only is it imperative to conduct more preclinical studies with female animal models to deepen our understanding of sex differences, but women are also underrepresented in clinical studies of AUD, despite being a group with significantly lower treatment completion rates compared to men [48]. Given these results, further studies should aim to elucidate the specific mechanisms OEA is targeting to modulate alcohol SA. Another limitation of this study is the use of the entire striatum and hippocampus of mice to perform qPCR analysis. In this sense, the future direction of this study will focus on protein expression analysis, in situ hybridization, or RNAScope to validate these gene expression findings.

## Figures and Tables

**Figure 1 ijms-25-09002-f001:**
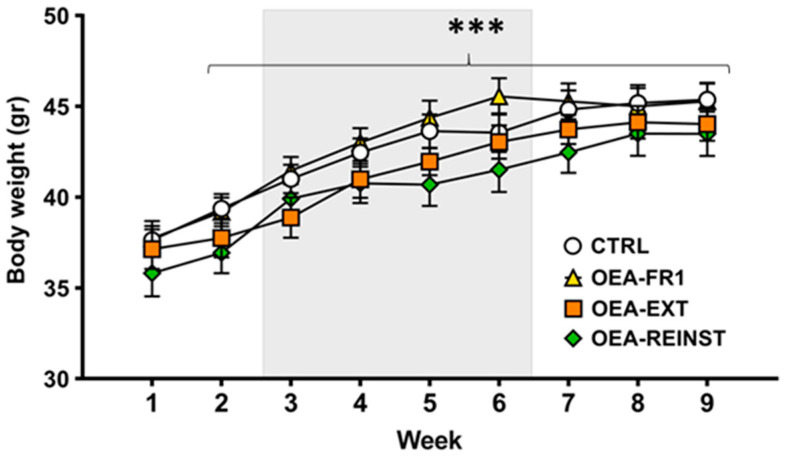
Bodyweight across the experiment. According to OEA treatment (10 mg/kg i.p): CTRL (n = 11, control group), OEA-FR1 (n = 11, 10 doses before each alcohol SA session), OEA-EXT (n = 12, 4–6 doses before each extinction session) and OEA-REINST (n = 12, a single dose before the cue-induced reinstatement test). Grey area represents alcohol 20% self-administration period (training and fixed-ratio 1). Data are represented as the mean (±SEM) bodyweight measured weekly. *** *p* < 0.001 significant difference with respect to Week 1, according to the repeated measure ANOVA followed by Bonferroni’s post hoc test.

**Figure 2 ijms-25-09002-f002:**
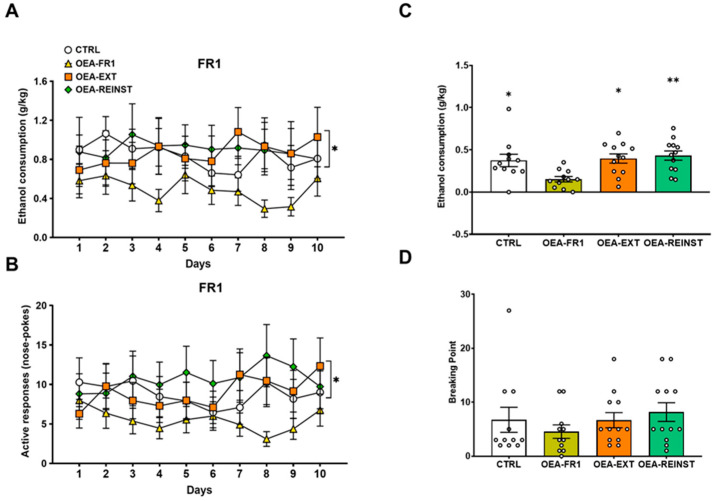
Effects of OEA treatment on oral alcohol self-administration. According to OEA treatment (10 mg/kg i.p): CTRL (n = 11, control group), OEA-FR1 (n = 11, 10 doses before each alcohol SA session), OEA-EXT (n = 12, 4-6 doses before each extinction session) and OEA-REINST (n = 12, a single dose before the cue-induced reinstatement test). The dots represent means and the vertical lines ± SEM of (**A**) 20% EtOH consumption during 10 days on a fixed-ratio 1 (FR1) (**B**) active nose poke responses during FR1 period. (**C**) EtOH consumption during the progressive ratio (PR) session, and (**D**) breaking point values during the PR session. * *p* < 0.05, ** *p* < 0.01 significant difference with respect to the OEA-FR1 group, according to the repeated measure ANOVA for FR1 period and one-way ANOVA for PR session, followed by Bonferroni’s post hoc test.

**Figure 3 ijms-25-09002-f003:**
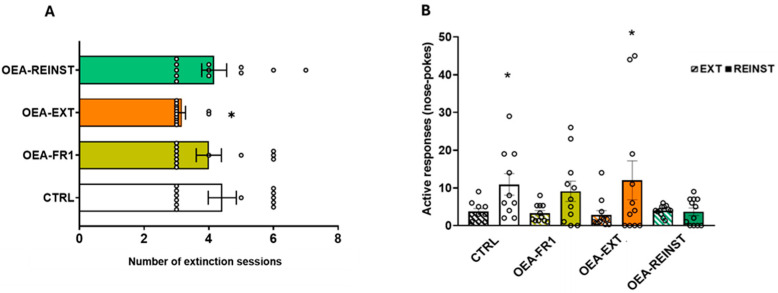
(**A**) Extinction. According to OEA treatment (10 mg/kg i.p): CTRL (n = 11, control group), OEA-FR1 (n = 11, 10 doses before each alcohol SA session), OEA-EXT (n = 12, 4–6 doses before each extinction session) and OEA-REINST (n = 12, a single dose before the cue-induced reinstatement test). The bars represent the total value (±S.E.M) of the number of sessions required for alcohol drug-seeking to be extinguished. * *p* < 0.05 significant difference with respect to the OEA-EXT group, according to the Kaplan–Meier test. (**B**) Effects of OEA treatment on cue-induced reinstatement of alcohol-seeking. Mean active nose pokes responses during the last three extinction sessions (represented by the cross hatched bars) and on the cue-induced reinstatement test. The columns represent means and the vertical lines ±SEM of active nose poke responses. * *p* < 0.05, significant difference with respect to EXT, according to the two-way ANOVA followed by Bonferroni’s post hoc test.

**Figure 4 ijms-25-09002-f004:**
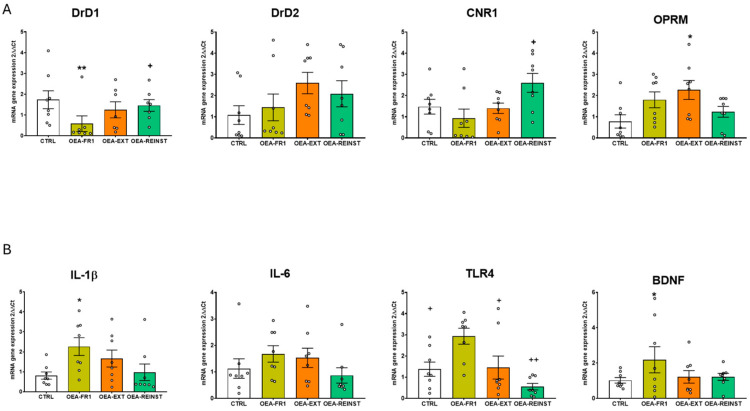
Effects of OEA administration on striatal gene expression of (**A**) relevant receptors involved in alcohol reinstatement such as: dopamine receptor DrD1, Drd2, cannabinoid receptor 1 (CNR1), mu-type opioid receptor (OPRM) (**B**) inflammatory genes, including interleukin IL-1β, IL-6, Toll-like receptor 4 (TLR4) and brain-derived neurotrophic factor (BDNF) after alcohol cue-induced reinstatement. According to OEA treatment (10 mg/kg i.p): CTRL (n = 8, control group), OEA-FR1 (n = 8, 10 doses before each alcohol SA session), OEA-EXT (n = 8, 4–6 doses before each extinction session) and OEA-REINST (n = 8, a single dose before the cue-induced reinstatement test). Bar graphs represent * *p* < 0.05, ** *p* < 0.01 significant difference with respect to the CTRL group; + *p* < 0.05, ++ *p* < 0.01 significant difference with respect to the OEA-FR1 group, according to the one-way ANOVA followed by Bonferroni’s post hoc test.

**Figure 5 ijms-25-09002-f005:**
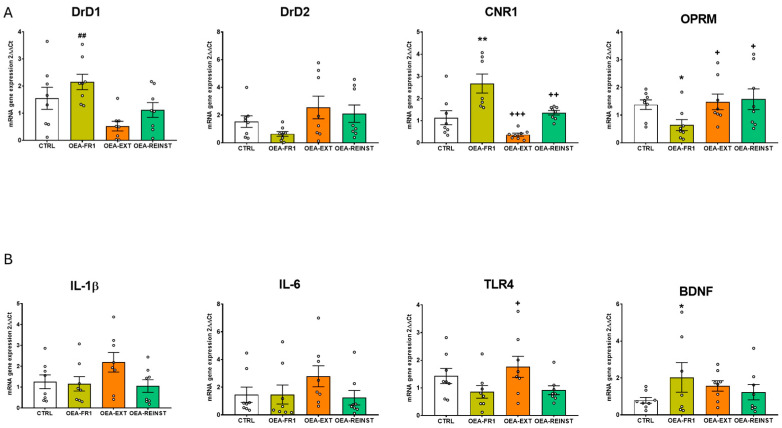
Effects of OEA administration on hippocampal gene expression of (**A**) relevant receptors involved in alcohol reinstatement such as: dopamine receptor DrD1, Drd2, cannabinoid receptor 1 (CNR1), mu-type opioid receptor (OPRM) (**B**) inflammatory genes, including interleukin IL-1β, IL-6, Toll-like receptor 4 (TLR4) and brain-derived neurotrophic factor (BDNF) after alcohol cue-induced reinstatement. According to OEA treatment (10 mg/kg i.p): CTRL (n = 8, control group), OEA-FR1 (n = 8, 10 doses before each alcohol SA session), OEA-EXT (n = 8, 4–6 doses before each extinction session) and OEA-REINST (n = 8, a single dose before the cue-induced reinstatement test). * *p* < 0.05, ** *p* < 0.01 significant differences with respect to the CTRL group; ## *p* < 0.01 significant difference with respect to the OEA-EXT group; + *p* < 0.05, ++ *p* < 0.01, +++ *p* < 0.001, significant difference with respect to the OEA-FR1 group, according to the one-way ANOVA followed by Bonferroni’s post hoc test.

**Figure 6 ijms-25-09002-f006:**
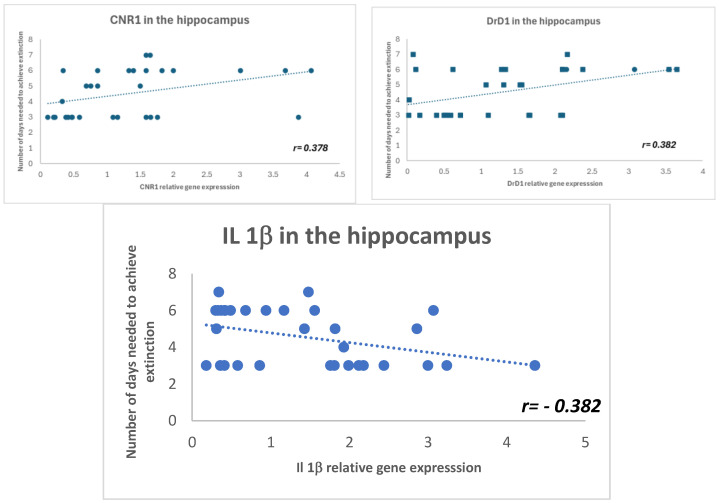
Pearson’s coefficient demonstrated that the expression of the DrD1 and CNR1 genes in the hippocampus positively correlates with the number of days required to extinguish oral ethanol self-administration. Conversely, IL-1β gene expression negatively correlates with the time needed to extinguish this operant behavior.

**Figure 7 ijms-25-09002-f007:**
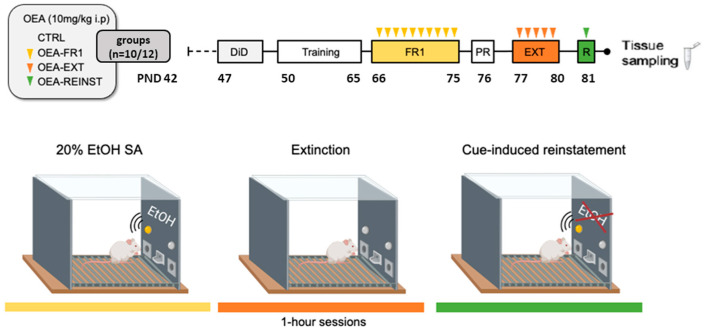
Experimental timeline. Late adolescent mice were randomly assigned into four experimental groups (n = 10/12) according to OEA treatment (10 mg/kg i.p): CTRL (control group), OEA-FR1 (10 doses before each alcohol SA session), OEA-EXT (4–6 doses before each extinction session) and OEA-REINST (a single dose before the cue-induced reinstatement test). Mice were habituated to alcohol during a four-day Drinking in the dark procedure (DID). After SA training, mice self-administered 20% (*v*/*v*) alcohol during 10 days on a fixed-ratio 1 (FR1). When the period of extinction for alcohol seeking resumed, mice were tested on a single cue-induced reinstatement session in which alcohol solution was not delivered. Tissue sampling was obtained after the SA procedure.

## Data Availability

The raw data supporting the conclusions of this article will be made available by the authors on request.

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
