# Peer review of "Region-Specific Gene Expression Changes Associated with Oleoylethanolamide-Induced Attenuation of Alcohol Self-Administration"

_ijms, 2024, doi:10.3390/ijms25169002_

Round 1

Reviewer 1 Report

Comments and Suggestions for Authors

In this manuscript, the authors, studied the effects of Oleoylethanolamide administration in alcohol-related behaviors. The focus of work is very interesting. The manuscript has several inaccuracies:

the introduction is very short and does not cover all the topics present in the manuscript.

- specify the effects of OEA in AUD patients, line 51

-  there is a lack of informations regarding the influence of the inflammatory state on alcohol-related behavior,

- lines 62-65 are not relevant,

- move the lines 74-77 after reference 13

In general, check the acronyms are always the same, for example DID and DiD.

Many acronyms are specified in the materials and methods, but appear first in the manuscript, making the results difficult to read. To better understand the results, the authors could specify the acronyms of the treatments in the figures.

The discussion is very long, the authors should shorten it.

Furthermore, gender differences in alcohol use disorder (AUD), treatment access, retention, outcomes, and long-term recovery are now well known. The authors do not address this issue in their work, add this point as a limitation of the study and specify the need to conduct other studies taking into account gender differences.

Author Response

Reviewer 1

The introduction is very short and does not cover all the topics present in the manuscript.

- specify the effects of OEA in AUD patients, line 51

Information about OEA in AUD patient has been added in the Introduction section:

Recently, a clinical study demonstrated that levels of NAEs, including OEA, are altered in patients with AUD. The plasma concentration of OEA was found to be elevated in alcohol-dependent patients during abstinence. Additionally, OEA concentrations were negatively correlated with the duration of alcohol abstinence [14, 15].

-  there is a lack of informations regarding the influence of the inflammatory state on alcohol-related behavior,

The following information has been included in the introduction section:

Additionally, alcohol consumption is associated with inflammatory states, significantly influencing alcohol-related behaviors. Chronic alcohol intake can exacerbate inflammation, creating a vicious cycle in which inflammation further promotes alcohol consumption. This relationship is mediated through various mechanisms, including the activation of Toll-like receptors (TLRs) and their downstream pathways in both peripheral cells (e.g., monocytes) and central cells (e.g., microglia) [5]. Furthermore, alcohol-induced gut dysbiosis results in the release of endotoxins from a compromised intestinal barrier, which circulate to the brain and other organs, triggering systemic and central inflammation. These inflammatory processes can alter neurochemical pathways, impacting mood, cognition, and reward processing [6, 7]. Consequently, individuals with heightened inflammatory states may experience increased alcohol cravings and consumption, thereby exacerbating the risk of AUD [7].

[5] Czerwińska-Błaszczyk, A., Pawlak, E., & Pawłowski, T. The significance of toll-like receptors in the neuroimmunologic background of alcohol dependence. Frontiers in psychiatry, 2022, 12, 797123. htpps://doi.org/10.3389/fpsyt.2021.797123

[6] Leclercq, S., Schwarz, M., Delzenne, N.M., Stärkel, P., & de Timary, P. Alterations of kynurenine pathway in alcohol use disorder and abstinence: A link with gut microbiota, peripheral inflammation and psychological symptoms. Translational Psychiatry, 2021, 11, 1–9. https://doi.org/ 10.1038/s41398-021-01610-5

[7] Kazmi, N., Wallen, G. R., Yang, L., Alkhatib, J., Schwandt, M. L., Feng, D., Gao, B., Diazgranados, N., Ramchandani, V. A., & Barb, J. J. An exploratory study of pro-inflammatory cytokines in individuals with alcohol use disorder: MCP-1 and IL-8 associated with alcohol consumption, sleep quality, anxiety, depression, and liver biomarkers. Frontiers in Psychiatry, 2022, 13, 931280. https://hoi.org/10.3389/fpsyt.2022.931280

Following the comments of reviewer 2, we have also added the following information to the introduction section:

Chronic alcohol exposure profoundly affects various systems in the human body, including the cardiovascular, gastrointestinal, and nervous systems. AUD is characterized by significant dysregulation in the brain's motivational circuits, driven by exaggerated incentive salience, habit formation, deficits in reward processing, increased stress, and impaired executive function [3-5]. These disruptions are primarily due to alcohol's effects on synaptic function, impacting multiple neurotransmitter systems such as serotonergic (5-HT), dopaminergic (DA), gamma-aminobutyric acid (GABA), glutamatergic (Glu), acetylcholinergic (ACh), and opioid systems [3-5].

Currently, there are several treatments available for AUD. Regulatory agencies have approved several medications, including disulfiram, naltrexone, and acamprosate [3].

Furthermore, there are several promising off-label pharmacotherapy options, such as nalmefene, baclofen, and topiramate. Despite the high prevalence, mortality, and economic impact of AUD, these medications remain under prescribed. Studies indicate that only a small number of individuals in need of treatment receive even a single prescription for any approved medication. This is likely due to barriers such as limited access, financial constraints, or inadequate health insurance coverage [3, 4].

[2] Yang, W., Singla, R., Maheshwari, O., Fontaine, C. J., & Gil-Mohapel, J. Alcohol use disorder: neurobiology and therapeutics. Biomedicines, 2022, 10(5), 1192. https://doi.org/10.3390/biomedicines10051192

[3] Prajapati, S. K., Bhaseen, S., Krishnamurthy, S., & Sahu, A. N. Neurochemical evidence of preclinical and clinical reports on target-based therapy in alcohol used disorder. Neurochemical Research, 2020, 45, 491-507. https://doi.org/10.1007/s11064-019-02944-9

[4] Liang, J., & Olsen, R. W. (2014). Alcohol use disorders and current pharmacological therapies: the role of GABAA receptors. Acta Pharmacologica Sinica, 35(8), 981-993. https://doi.org/10.1038/aps.2014.50

- lines 62-65 are not relevant,

We have removed part of lines 62-65.

- move the lines 74-77 after reference 13

Lines 74-77 have been moved after the reference [13], which is number 18 after the revision.

In general, check the acronyms are always the same, for example DID and DiD.

According to the reviewer comment, the acronyms in the manuscript have been reviewed.

Many acronyms are specified in the materials and methods, but appear first in the manuscript, making the results difficult to read. To better understand the results, the authors could specify the acronyms of the treatments in the figures.

Thank you very much for your comment. We have now ensured that the acronyms for the treatments and the various aspects analyzed in the graphs are clearly described in each of the figures.

The discussion is very long, the authors should shorten it.

Discussion has been shortened from 1647 to 1475 words. Although we tried to made a more important reduction of the discussion, the part devoted to gene expression continue being very extensive due to the high number of genes studied and the new information added regarding the correlation study.

Furthermore, gender differences in alcohol use disorder (AUD), treatment access, retention, outcomes, and long-term recovery are now well known. The authors do not address this issue in their work, add this point as a limitation of the study and specify the need to conduct other studies taking into account gender differences.

The following information has been added to the conclusion section:

A notable limitation of this study is the exclusive use of male mice. It is well established that there are significant gender differences in AUD, with complex interactions among genetic, epigenetic, hormonal, and environmental factors contributing to these disparities [46]. Furthermore, aspects such as the higher prevalence of mental disorders, including depression, associated with AUD in women, demand additional studies to develop treatments tailored specifically for this demographic [47]. Not only is imperative to conduct more preclinical studies with female animal models to deepen our understanding of sex differences, but women are also underrepresented in clinical studies of AUD, despite being a group with significantly lower treatment completion rates compared to men [48].

[46] Maddern XJ, Ursich LT, Bailey G, Pearl A, Anversa RG, Lawrence AJ, Walker LC. Sex Differences in Alcohol Use: Is It All About Hormones? Endocrinology. 2024 Jul 26;165(9):bqae088. doi: 10.1210/endocr/bqae088. PMID: 39018449.

[47] McHugh RK, Weiss RD. Alcohol Use Disorder and Depressive Disorders. Alcohol Res. 2019 Jan 1;40(1):arcr.v40.1.01. doi: 10.35946/arcr.v40.1.01. PMID: 31649834; PMCID: PMC6799954.

[48] Delk J, Bensley K, Ye Y, Subbaraman MS, Phillips AZ, Karriker-Jaffe KJ, Mulia N. Intersectional disparities in outpatient alcohol treatment completion by gender and race and ethnicity. Alcohol Clin Exp Res (Hoboken). 2024 Feb;48(2):389-399. doi: 10.1111/acer.15243. Epub 2024 Feb 1. PMID: 38300125; PMCID: PMC10922739.

Reviewer 2 Report

Comments and Suggestions for Authors

Author Response

REVIEWER 2

Abstract:

Authors must define the abbreviation of important genes used in this study

These abbreviations have been introduced in the manuscript:

  • Dopamine receptor D1 and D2 (DrD1, DrD2)
  • Cannabinoid receptor 1 (CNR1)
  • Opioid receptor mu-1 (OPRM)
  • Interleukin-6 (IL-6)
  • Interleukin-1β (IL-1β)
  • Toll-like receptor-4 (TLR4)
  • Drain-derived neurotrophic factor (BDNF)

Introduction:

Introduction is too short to follow, and author must elaborate at least one para

regarding neurochemical changes and therapeutics intervention in AUD.

https://www.ncbi.nlm.nih.gov/pmc/articles/PMC9139063/

https://pubmed.ncbi.nlm.nih.gov/31898084/

https://www.nature.com/articles/aps201450

According to the reviewer comment, and using the references proposed, two paragraphs were added in the introduction section:

Chronic alcohol exposure profoundly affects various systems in the human body, including the cardiovascular, gastrointestinal, and nervous systems. AUD is characterized by significant dysregulation in the brain's motivational circuits, driven by exaggerated incentive salience, habit formation, deficits in reward processing, increased stress, and impaired executive function [2-5]. These disruptions are primarily due to alcohol's effects on synaptic function, impacting multiple neurotransmitter systems such as serotonergic (5-HT), dopaminergic (DA), gamma-aminobutyric acid (GABA), glutamatergic (Glu), acetylcholinergic (ACh), and opioid systems [2-5].

Currently, there are several treatments available for AUD. Regulatory agencies have approved several medications, including disulfiram, naltrexone, and acamprosate [3].

Furthermore, there are several promising off-label pharmacotherapy options, such as nalmefene, baclofen, and topiramate. Despite the high prevalence, mortality, and economic impact of AUD, these medications remain under prescribed. Studies indicate that only a small number of individuals in need of treatment receive even a single prescription for any approved medication. This is likely due to barriers such as limited access, financial constraints, or inadequate health insurance coverage [3, 4].

[2] Yang, W., Singla, R., Maheshwari, O., Fontaine, C. J., & Gil-Mohapel, J. Alcohol use disorder: neurobiology and therapeutics. Biomedicines, 2022, 10(5), 1192. https://doi.org/10.3390/biomedicines10051192

[3] Prajapati, S. K., Bhaseen, S., Krishnamurthy, S., & Sahu, A. N. Neurochemical evidence of preclinical and clinical reports on target-based therapy in alcohol used disorder. Neurochemical Research, 2020, 45, 491-507. https://doi.org/10.1007/s11064-019-02944-9

[4] Liang, J., & Olsen, R. W. (2014). Alcohol use disorders and current pharmacological therapies: the role of GABAA receptors. Acta Pharmacologica Sinica, 35(8), 981-993. https://doi.org/10.1038/aps.2014.50

In addition, this information has also been added to the introduction section:

Additionally, alcohol consumption is associated with inflammatory states, significantly influencing alcohol-related behaviors. Chronic alcohol intake can exacerbate inflammation, creating a vicious cycle in which inflammation further promotes alcohol consumption. This relationship is mediated through various mechanisms, including the activation of Toll-like receptors (TLRs) and their downstream pathways in both peripheral cells (e.g., monocytes) and central cells (e.g., microglia) [5]. Furthermore, alcohol-induced gut dysbiosis results in the release of endotoxins from a compromised intestinal barrier, which circulate to the brain and other organs, triggering systemic and central inflammation. These inflammatory processes can alter neurochemical pathways, impacting mood, cognition, and reward processing [6, 7]. Consequently, individuals with heightened inflammatory states may experience increased alcohol cravings and consumption, thereby exacerbating the risk of AUD [7].

 [5] CzerwiÅ„ska-BÅ‚aszczyk, A., Pawlak, E., & PawÅ‚owski, T. The significance of toll-like receptors in the neuroimmunologic background of alcohol dependence. Frontiers in psychiatry, 2022, 12, 797123. htpps://doi.org/10.3389/fpsyt.2021.797123

[6] Leclercq, S., Schwarz, M., Delzenne, N.M., Stärkel, P., & de Timary, P. Alterations of kynurenine pathway in alcohol use disorder and abstinence: A link with gut microbiota, peripheral inflammation and psychological symptoms. Translational Psychiatry, 2021, 11, 1–9. https://doi.org/ 10.1038/s41398-021-01610-5

[7] Kazmi, N., Wallen, G. R., Yang, L., Alkhatib, J., Schwandt, M. L., Feng, D., Gao, B., Diazgranados, N., Ramchandani, V. A., & Barb, J. J. An exploratory study of pro-inflammatory cytokines in individuals with alcohol use disorder: MCP-1 and IL-8 associated with alcohol consumption, sleep quality, anxiety, depression, and liver biomarkers. Frontiers in Psychiatry, 2022, 13, 931280. https://hoi.org/10.3389/fpsyt.2022.931280

Explain how OEA could be different from existing: just in brief.

Oleoylethanolamide (OEA) is an endogenous lipid with anti-inflammatory properties that has shown potential in modulating alcohol-related behavior through interaction with PPAR-α receptors [17]. This differs from the mechanisms of action of the current FDA-approved treatments for alcohol use disorder (AUD). The OEA primarily acts through the activation of PPAR-α receptors, modulating the expression of genes related to inflammation and energy homeostasis [17]. Disulfiram treatment, inhibits the enzyme acetaldehyde dehydrogenase, causing an accumulation of acetaldehyde when alcohol is consumed, leading to unpleasant effects [2]. Naltrexone acts as an antagonist of opioid receptors, reducing the rewarding effects of alcohol [2]. And the last one, acamprosate, modulates the activity of the glutamatergic system, helping to restore the neurochemical balance disrupted by chronic alcohol consumption [2]. These differences highlight the unique potential of OEA in treating alcohol use disorder, particularly concerning the modulation of inflammation and the regulation of energy homeostasis.

[2] Yang, W., Singla, R., Maheshwari, O., Fontaine, C. J., & Gil-Mohapel, J. Alcohol use disorder: neurobiology and therapeutics. Biomedicines, 2022, 10(5), 1192. https://doi.org/10.3390/biomedicines10051192

[17] Guzmán, M., Verme, J. L., Fu, J., Oveisi, F., Blázquez, C., & Piomelli, D. Oleoylethanolamide stimulates lipolysis by activating the nuclear receptor peroxisome proliferator-activated receptor α (PPAR-α). J Biol Chem, 2004, 279(27), 27849-27854. https://doi.org/10.1074/jbc.M404087200

Methodology:

I would be interested to observe how these gene expressions vary or replicate with different rodent models of AUD.

Chronic alcohol consumption (as studied in mice maintained on water containing 10% v/v alcohol for 5 months) induces neuroinflammatory molecules, such as genes and microRNAs, associated with the activation of the TLR4 immune response in mice (Ureña-Peralta et al., 2018). Other researchers have found that chronic alcohol consumption (using the two-bottle choice and drinking in the dark procedures) is linked to a highly over-represented dopamine-related network in the prefrontal cortex (e.g., DrD1, DrD2). This effect is like that seen in lipopolysaccharide-induced neuroimmune activation, suggesting a connection between immune response, ethanol intake, and dopamine signaling in the murine brain (Osterndorff-Kahanek et al., 2013). Additionally, blocking CNR1 using antagonists significantly reduced alcohol-induced pyroptosis (triggered by 95% alcohol vapor exposure for 30 days) and attenuated the inflammatory response in mice (Zhang et al., 2020).

Regarding OPRM1, in a mouse model using the drinking in the dark (DID) protocol for 4 weeks, female mice exhibited higher expression of this gene (Viudez-Martínez et al., 2019).

Finally, chronic alcohol intoxication (in rats exposed to alcohol vapor for 4 weeks) decreases BDNF mRNA expression in the rat hippocampus and the supraoptic nucleus of the hypothalamus, which is associated with an up-regulation of trkB mRNA expression (Tapia-Arancibia et al., 2001).

Ureña-Peralta JR, Alfonso-Loeches S, Cuesta-Diaz CM, García-García F, Guerri C. Deep sequencing and miRNA profiles in alcohol-induced neuroinflammation and the TLR4 response in mice cerebral cortex. Sci Rep. 2018 Oct 29;8(1):15913. doi: 10.1038/s41598-018-34277-y. PMID: 30374194; PMCID: PMC6206094.

Osterndorff-Kahanek E, Ponomarev I, Blednov YA, Harris RA. Gene expression in brain and liver produced by three different regimens of alcohol consumption in mice: comparison with immune activation. PLoS One. 2013;8(3):e59870. doi: 10.1371/journal.pone.0059870. Epub 2013 Mar 29. PMID: 23555817; PMCID: PMC3612084.

Zhang D, Liu X, Dong X, Zhu R, Jiang J, Ye Y, Jiang Y. Cannabinoid 1 Receptor Antagonists Play a Neuroprotective Role in Chronic Alcoholic Hippocampal Injury Related to Pyroptosis Pathway. Alcohol Clin Exp Res. 2020 Aug;44(8):1585-1597. doi: 10.1111/acer.14391. Epub 2020 Jul 1. PMID: 32524615.

Viudez-Martínez A, García-Gutiérrez MS, Manzanares J. Gender differences in the effects of cannabidiol on ethanol binge drinking in mice. Addict Biol. 2020 May;25(3):e12765. doi: 10.1111/adb.12765. Epub 2019 May 9. PMID: 31074060.

Tapia-Arancibia L, Rage F, Givalois L, Dingeon P, Arancibia S, Beaugé F. Effects of alcohol on brain-derived neurotrophic factor mRNA expression in discrete regions of the rat hippocampus and hypothalamus. J Neurosci Res. 2001 Jan 15;63(2):200-8. doi: 10.1002/1097-4547(20010115)63:2<200::AID-JNR1012>3.0.CO;2-Q. PMID: 11169630.

Results:

It is hard to conclude due to the smaller sample size.

We agree with the reviewer that a larger sample size would enhance the strength of the results. However, we used a sample size typical for the experimental procedures employed in this study. For gene expression studies, a sample size of N=8 is commonly used. In contrast, for ethanol oral self-administration studies, sample sizes typically range between 10 and 16 (for example, see Reguilon et al., 2023 and 2022; Blanco-Gandia et al., 2021). The ethics committee restricted us from increasing the sample size. The GPower simulation determined the maximum number of animals per group. For this experimental design, the GPower simulation allowed for 48 mice across 4 experimental groups (n=12 per group).

Reguilón MD, Ferrer-Pérez C, Manzanedo C, Miñarro J, Rodríguez-Arias M. Voluntary wheel running during adolescence prevents the increase in ethanol intake induced by social defeat in male mice. Psychopharmacology (Berl). 2023 Sep 22. doi: 10.1007/s00213-023-06461-0. Epub ahead of print. PMID: 37736785.

Reguilón MD, Ballestín R, Miñarro J, Rodríguez-Arias M. Resilience to social defeat stress in adolescent male mice. Prog Neuropsychopharmacol Biol Psychiatry. 2022 Dec 20;119:110591. doi: 10.1016/j.pnpbp.2022.110591. Epub 2022 Jun 10. PMID: 35697171.

Blanco-Gandía MDC, Ródenas-González F, Pascual M, Reguilón MD, Guerri C, Miñarro J, Rodríguez-Arias M. Ketogenic Diet Decreases Alcohol Intake in Adult Male Mice. Nutrients. 2021 Jun 24;13(7):2167. doi: 10.3390/nu13072167. PMID: 34202492; PMCID: PMC8308435.

I don’t see blood ethanol measurement results which is important to correlate it with reward and alcohol preference.

We agree with the reviewer on the importance of measuring blood ethanol concentration (BEC) to better understand our results. Although we did not measure BEC in this study, we have measured it in a previous study using mice of the same strain (OF1) and age. In that study, the results showed that for mice with an ethanol intake of 1.4 g/kg, the BEC was 25 mg/dl. For those consuming 2.4 g/kg, the BEC was 40 mg/dl (Blanco-Gandia et al., 2018).

In the present study, ethanol intake ranged from 1.6 to 3.4 g/kg, so we can infer that the lowest BEC was approximately 25 mg/dl. During the self-administration FR1 phase, ethanol intake was consistently below 1 g/kg, suggesting that the corresponding BEC would be lower.

We acknowledge this limitation and are working on implementing a method to measure BEC. This limitation has been noted in the discussion section.

A potential limitation of the study is the absence of blood ethanol concentration measurements, which could have been correlated with ethanol-induced reinforcement.

Blanco-Gandía MC, Miñarro J, Aguilar MA, Rodríguez-Arias M. Increased ethanol consumption after interruption of fat bingeing. PLoS One. 2018 Mar 28;13(3):e0194431. doi: 10.1371/journal.pone.0194431. PMID: 29590149; PMCID: PMC5874030.

I would recommend correlation between phenotype and gene expression in different brain regions.

Following the reviewer's suggestion, we conducted a correlation analysis between gene expression results and ethanol intake during the DID period and oral ethanol self-administration (FR1), the number of days needed for achieve extinction and the number of effective responses in the progressive ratio and the reinstatement test. The results are presented below. This new information has been added to the Results section and discussed accordingly in the Discussion.

Results

We performed a Pearson correlation analysis between the total ethanol intake during the DID and FR1 procedures and the expression of the studied genes in the hippocampus and striatum. No significant correlations were found. Similarly, no significant correlations were observed between the number of effective responses during the progressive ratio or reinstatement test. However, we identified a significant positive Pearson correlation between the number of days required to achieve extinction and the hippocampal expression of DrD1 (r = 0.382, p < 0.031) and CNR1 (r = 0.378, p < 0.036). This suggests that higher gene expression levels are associated with an increased number of days needed to extinguish operant ethanol self-administration. Additionally, a significant negative correlation was observed with IL-1β expression (r = -0.382, p < 0.031), indicating that higher IL-1β gene expression is associated with fewer days required to achieve extinction.

Figure 6: Expression of the DrD1 and CNR1 genes in the hippocampus positively correlates with the number of days required to extinguish oral ethanol self-administration. Conversely, IL-1β gene expression negatively correlates with the time needed to extinguish this operant behavior.

Discussion and conclusion:

Discussion is good however, inclusion of blood ethanol measurement with current and using another moder would give more robust findings

This limitation has been added to the discussion section.

Reviewer 3 Report

Comments and Suggestions for Authors

Macarena et al. investigated the effects of oleoylethanolamide (OEA) on alcohol self-administration (ASA) and the expression of various genes in the striatum and hippocampus. While the overall conclusion is robust, the authors should include additional supporting data and more detailed experimental information in the figure legends.

1. Please use the official gene names, such as Drd1 and Drd2. Names such as DrD1 and DrD2 are not proper.

2. In all the legends of all figures, please provide the n value and the way of statistics. The authors need to indicate the comparisons in a clearer way, like in Fig. 2 C & D, 3, 4 and 5.  

3. For QPCR experiments, I’d like to see how the primers work. The authors need to present some gels which can show the primers work well.

4. The data of region-specific gene expression is not strong enough. More data is required to support the conclusion. Here are several: (a) check whether the protein levels of those genes have also changed. (b) Do some in situ hybridization or RNAScope to validate the QPCR findings.  

5. The authors should cite more newly publications.

Comments on the Quality of English Language

Need to polish it and pay some attension on the format.

Author Response

Reviewer 3

While the overall conclusion is robust, the authors should include additional supporting data and more detailed experimental information in the figure legends.

This change has been done

  1. Please use the official gene names, such as Drd1 and Drd2. Names such as DrD1 and DrD2 are not proper.

 This change has been done

  1. In all the legends of all figures, please provide the n value and the way of statistics. The authors need to indicate the comparisons in a clearer way, like in Fig. 2 C & D, 3, 4 and 5. 

Statistic methods used for each of comparisons are described in material and methods and also the values are specified in the results methods. We have added the n of each group in the legend of the figures.

  1. For QPCR experiments, I’d like to see how the primers work. The authors need to present some gels which can show the primers work well.

We're sorry, but we cannot provide the information requested by the reviewer. Melt curve analysis is always used in our qPCR studies as a diagnostic tool, to corroborate that the qPCR assays have produced a single and specific qPCR product. In addition, each qPCR assay is performed with triplicate samples and amplification plots are also used to validate good PCR reactions.

Here some examples of our previous papers in which the same procedure has been employed:

Immunology and Cell Biology (2011) 89, 716–727.  doi:10.1038/icb.2010.163

Toxicology Volume (2013) 311, 27-34. doi.org/10.1016/j.tox.2013.03.001

Cell Death and Disease (2014) 5, e1066. doi:10.1038/cddis.2014.46

Brain, Behavior, and Immunity (2016) 53, 159-171. doi.org/10.1016/j.bbi.2015.12.006

Brain Pathology (2021) 31, 174–188. doi:10.1111/bpa.12896

Int. J. Mol. Sci. (2021) 22, 8438. doi.org/10.3390/ijms22168438

CNS Neurosci Ther. (2023) 00:1–14. doi: 10.1111/cns.14326

  1. The data of region-specific gene expression is not strong enough. More data is required to support the conclusion. Here are several: (a) check whether the protein levels of those genes have also changed. (b) Do some in situ hybridization or RNAScope to validate the QPCR findings.

 Although it would have been great to carry out the studies proposed by the reviewer, we do not have the samples to do so. This would mean having to redo the entire study to obtain more samples, which is impossible. We will keep this in mind for future publications.

We would like to point out, as mentioned in the previous point, that gene expression data are generally considered very valuable and provide different information compared to the techniques proposed by the reviewer. The importance and validity of gene expression studies are demonstrated by the high number of publications in which we have exclusively presented this type of data in relation to the consumption of different drugs (see some examples in the previous point)

González-Portilla M, Mellado S, Montagud-Romero S, Rodríguez de Fonseca F, Pascual M, Rodríguez-Arias M. Oleoylethanolamide attenuates cocaine-primed reinstatement and alters dopaminergic gene expression in the striatum. Behav Brain Funct. 2023 May 24;19(1):8. doi: 10.1186/s12993-023-00210-1. PMID: 37226219; PMCID: PMC10207629.

Blanco-Gandía MDC, Ródenas-González F, Pascual M, Reguilón MD, Guerri C, Miñarro J, Rodríguez-Arias M. Ketogenic Diet Decreases Alcohol Intake in Adult Male Mice. Nutrients. 2021 Jun 24;13(7):2167. doi: 10.3390/nu13072167. PMID: 34202492; PMCID: PMC8308435.

Ródenas-González F, Blanco-Gandía MDC, Pascual M, Molari I, Guerri C, López JM, Rodríguez-Arias M. A limited and intermittent access to a high-fat diet modulates the effects of cocaine-induced reinstatement in the conditioned place preference in male and female mice. Psychopharmacology (Berl). 2021 Aug;238(8):2091-2103. doi: 10.1007/s00213-021-05834-7. Epub 2021 Mar 31. PMID: 33786639.

Gasparyan A, Navarrete F, Rodríguez-Arias M, Miñarro J, Manzanares J. Cannabidiol Modulates Behavioural and Gene Expression Alterations Induced by Spontaneous Cocaine Withdrawal. Neurotherapeutics. 2021 Jan;18(1):615-623. doi: 10.1007/s13311-020-00976-6. Epub 2020 Nov 23. PMID: 33230690; PMCID: PMC8116402.

  1. The authors should cite more newly publications.

Several papers of 2024 have been added to the manuscript

Self-citation issue:

Editor: During the technical check of your manuscript, we noticed that a high proportion of the cited references belong to you or your co-authors Refs. 5, 8, 10, 11, 14, 15, 17, 24, 33, 34, 35, 37, 42, 43 which is a self-citation rate of about 29.79%. (It should be under than 20%).

We understand the requirement of limiting the number of self-citations, but it should be noted that the study of the effects of OEA on alcohol consumption has been conducted by a very limited number of researchers, including the authors of this manuscript. Therefore, the use of self-citations is unavoidable. However, following the editor's comment, we have removed 5 self-citations. This, along with the increase in the number of references in the revised manuscript, reduces the percentage of self-citations to only 22,9%.

Round 2

Reviewer 1 Report

Comments and Suggestions for Authors

Thanks for following my suggestions. The manuscript has improved a lot.

Author Response

Thank you for the reviewer's comments, all of them have been included in the manucript.

Reviewer 2 Report

Comments and Suggestions for Authors

Authors have addressed all the comments.

Author Response

(The authors gave the same response as above.)

Reviewer 3 Report

Comments and Suggestions for Authors

More expreiments are rquired.

Comments on the Quality of English Language

Pay more attend to the format.

Author Response

Thank you for the reviewer's comments. We have included the official gene names in mus musculus in the entire manuscript. In addition, the n values and the statistics including type of ANOVA and post hoc test have also been included in the figure legends. Regarding the comments 3 and 4, we have included some sentences in the M&M and Conclusions sections.